# Requirements for Processing High-Strength AlZnMgCu Alloys with PBF-LB/M to Achieve Crack-Free and Dense Parts

**DOI:** 10.3390/ma14237190

**Published:** 2021-11-25

**Authors:** Steffen Heiland, Benjamin Milkereit, Kay-Peter Hoyer, Evgeny Zhuravlev, Olaf Kessler, Mirko Schaper

**Affiliations:** 1Chair of Materials Science, Paderborn University, 33098 Paderborn, Germany; hoyer@lwk.upb.de (K.-P.H.); schaper@lwk.uni-paderborn.de (M.S.); 2Paderborn Institute for Additive Fabrication (PIAF), 33100 Paderborn, Germany; 3Chair of Materials Science, University of Rostock, 18055 Rostock, Germany; benjamin.milkereit@uni-rostock.de (B.M.); evgeny.zhuravlev@uni-rostock.de (E.Z.); olaf.kessler@uni-rostock.de (O.K.); 4Competence Centre °CALOR, Department Life, Light & Matter, University of Rostock, 18055 Rostock, Germany

**Keywords:** grain refinement, crack reduction, laser beam melting, aluminum alloy, titanium carbide, nanoparticle, PBF-LB/M

## Abstract

Processing aluminum alloys employing powder bed fusion of metals (PBF-LB/M) is becoming more attractive for the industry, especially if lightweight applications are needed. Unfortunately, high-strength aluminum alloys such as AA7075 are prone to hot cracking during PBF-LB/M, as well as welding. Both a large solidification range promoted by the alloying elements zinc and copper and a high thermal gradient accompanied with the manufacturing process conditions lead to or favor hot cracking. In the present study, a simple method for modifying the powder surface with titanium carbide nanoparticles (NPs) as a nucleating agent is aimed. The effect on the microstructure with different amounts of the nucleating agent is shown. For the aluminum alloy 7075 with 2.5 ma% titanium carbide nanoparticles, manufactured via PBF-LB/M, crack-free samples with a refined microstructure having no discernible melt pool boundaries and columnar grains are observed. After using a two-step ageing heat treatment, ultimate tensile strengths up to 465 MPa and an 8.9% elongation at break are achieved. Furthermore, it is demonstrated that not all nanoparticles used remain in the melt pool during PBF-LB/M.

## 1. Introduction

High-strength aluminum alloys such as alloys of the 7xxx series are, due to their low density and good mechanical properties (excellent strength-to-weight ratio), one of the most important construction materials, in particular in the aerospace and aeronautic industry [1,2]. However, the hot crack susceptibility of these alloys is one major challenge that experts have been facing for decades, and they have been looking for the mechanism to solve this challenge [3,4,5]. Moreover, poor flowability [6], high reflectivity, as well as high thermal conductivity [7] are further challenges limiting the deployment of these materials in laser-based powder bed fusion of metals (PBF-LB/M) [8,9], also known as laser beam melting (LBM).

The phenomenon of occurring hot cracks is well-known. Regarding this, numerous studies have been conducted for aluminum alloys with the propensity to hot cracking, mainly for alloys of the 2xxx, 5xxx, 6xxx, and 7xxx series. In particular, aluminum alloys with the alloying elements copper (Cu) and magnesium (Mg) tend to have a higher crack sensitivity [10,11,12,13]. From the material aspect, the alloy composition itself can also be changed by evaporation of volatile alloying elements (e.g., zinc (Zn)) during PBF-LB/M [4]. This can lead to a concentration quantity and ratio of alloying elements, resulting in a large solidification range and solidification shrinkage [14], which can affect the generation of hot tears. In addition, the laser beam melting production conditions such as rapid cooling rates (10^3^ K/s–10^5^ K/s [15]) during solidification and heat input (intensity and source) have also been considered concerning hot crack formation [12,16]. Different approaches to process hard-to-weld Al-Zn-Mg-Cu alloys (7xxx) by welding and additive manufacturing, respectively, have been conducted. Investigations regarding the PBF-LB/M process parameters and their influence on the relative density and crack formation for the aluminum alloy 7075 were performed by Kaufmann et al. [17] and Stopyra et al. [16]. Both investigations show that with increasing laser power, a relative density of more than 99% can be achieved in AA7075 by applying high scan speeds; nonetheless, hot cracks still occurred.

Typically, the microstructure after PBF-LB/M exhibits dendritic grains which are semicircularly arranged for each melt pool. Generating small equiaxed grains during solidification decreases the hot tearing susceptibility [18]. Grain refining can be promoted by modifying the chemical composition of the basic alloy system thereof resulting in precipitates, which act as nuclei [19]. But also, stable solid particles are applied to cause heterogenic nucleation. In recent years, the application of micro- and/or nanoparticles (NPs) of a predefined nucleation agent or alloying elements for materials in conjunction with welding and additive manufacturing has become increasingly popular.

Montero-Sistiaga et al., attained a reduction of hot cracks and a relative density of 98.8% after adding at least 3 ma% silicon (Si) to the aluminum alloy 7075. The melting temperature, as well as the solidification range, decreased with increasing silicon content based on the formation of a eutectic [20]. Martin et al., processed 7075 with 1 vol% hydrogen-stabilized zirconium nanoparticles via PBF-LB/M resulting in crack-free specimens. The microstructure exhibits equiaxed grains with sizes of roughly 5 µm [21]. Grain refinement with an average grain size of 1.94 µm in AA7075 and samples without cracks after PBF-LB/M were achieved with the addition of 4 ma% titanium nitride (TiN) nanoparticles [22]. Transferred from casting with Al-Ti-C refiners [23,24], titanium carbide (TiC) particles are also used as a direct nucleant since they have a high melting point (3140 °C [25]) and a low lattice mismatch to the aluminum matrix, making them a suitable grain refiner [23,26]. For successful crack-free arc welding of the aluminum alloy 7075, Sokoluk et al., applied filler containing 1.7 vol% TiC nanoparticles [26]. The aluminum alloy AlSi10Mg was reinforced with 5 ma% TiC nanoparticles and manufactured with PBF-LB/M by Gu et al. [27] resulting in a hardness increase of about 25%. A TiC-modified AA5024 containing scandium (Sc) and zirconium (Zr), well-known as effective grain refiners, was investigated for laser metal deposition and revealed no appreciable effects in increasing strength or hardness [28]. However, for AA7075 reinforced with 4 ma% TiB_2_ µm-particles and processed by using laser metal deposition, an increase in hardness of more than 50% and a decrease of the grain size by 32% were observed [29]. It can be observed that the vast majority of published studies concerning additive manufacturing of aluminum alloys and changes in the microstructure mainly focus on 6xxx aluminum alloys, two alloying systems like Al-Si alloys with very low hot cracking susceptibility and high-strength aluminum alloys of the 2xxx series [3,30], which can be fabricated free of microcracks and imperfections by powder bed fusion without foreign particles initiating grain refinement [31].

Surface deposition of basis powder with additives, often ceramic or metallic particles with micro- and/or nano-scale sizes can be used for enhancing the absorbency to increase the thermal conductivity and thereof, the processability using additive manufacturing procedures [32]. Furthermore, such particles promote grain refinement to enhance the PBF-LB/M processability on the material aspect, e.g., in improving the mechanical properties [6,33].

Dry mixing with [34] and without [34,35] the previous use of an ultrasonic vibrator, ball milling [15,28,30], satelliting technique [36,37], or fluidized bed system [38] are appropriate techniques, with advantages and disadvantages of each, to join particulate materials (chemical elements, ceramics) on the particle surface of the base powder material.

The present study investigates the effect of TiC nanoparticles applied with an easy-to-perform modification procedure on the microstructure and mechanical properties of a hard-to-weld high-strength AlZnMgCu alloy processed via PBF-LB/M. An objective of this study is to answer the question regarding the need for a sufficient quantity of a nucleation agent to achieve crack-free samples as well as the correlation between resulting grain refinement by using the average grain size and scanning strategy. Furthermore, and according to the authors’ best knowledge, for the first time, the chemical composition of a nanoparticle-modified powder is investigated after several powder processing stages, as well as the melt pool ejection, which indicates the remaining nanoparticles during powder handling and additive processing.

## 2. Materials and Methods

### 2.1. Mechanical Decoration of the Powder

Starting material is the gas atomized spherical high-strength aluminum alloy Al-5.0Zn-1.3Mg-1.5Cu (in ma%) (Engineering for You GmbH) with a particle size distribution of d_10_ = 14.17 µm, d_50_ = 34.64 µm, d_90_ = 66.80 µm. The chemical composition of the virgin starting powder summarized in Table 1 was determined by X-ray fluorescence spectroscopy. Due to the chemical composition close to EN AW-7075 according to DIN EN 573-3 [39], the alloy investigated is denoted here as AA7075. For surface decoration of the basic powder and serving as a nuclei agent, TiC nanoparticles (IoLiTec Ionic Liquids Technologies GmbH, Germany) with a purity of 99% (manufacturer information) and a mean diameter of 40.3 nm were applied. Mechanically mixing was performed with the roller mill RM01s (Zoz GmbH, Germany). A vessel with a maximum capacity of one liter and the ability to evacuate the atmosphere was filled with AA7075 powder and 0.00 ma%/0.50 ma%/1.00 ma%/1.75 ma%/2.50 ma% TiC nanopowder as well as stainless steel balls with a diameter of 9.5 mm. Using a powder-to-ball weight ratio of 1:6, 350 g of mechanically decorated powder was produced in each lot. After the evacuation of the container, the powders were mechanically mixed for 30 min, applying a rotation speed of 120 rpm. Before laser beam melting and between each build job, the modified powder was sifted with a mesh size of 90 µm (Mini Sonic Screen MSS 150, assonic Dorsteiner Siebtechnik GmbH, Germany) and subsequently dried with a vacuum dryer (AM1000, AMproved GmbH, Germany) until relative moisture of ≤5% was obtained. For determining the moisture, an Hytelog-USB humidity sensor (B + B Thermo-Technik GmbH, Germany) was utilized.

### 2.2. Laser Beam Melting of Specimens

For powder bed fusion, an SLM 250^HL^ system (SLM Solutions Group AG, Germany), equipped with a YLM-400-WC Laser (IPG Photonics, USA) having a maximum laser power of 400 W and a wavelength of 1.07 µm was operated in the continuous mode. The applied laser spot diameter is ≈90 µm (focus shift of +1 mm) with a Gaussian beam profile. To avoid oxidation, particularly for processing aluminum one remarkable challenge [7], the build chamber was filled with argon 4.6 (99.996 vol%) using a pressure of 75 mbar. The oxygen level during the build process was below 0.04 vol%. The build plate, made of an 5xxx aluminum alloy, was preheated to 200 °C affecting the thermal gradient. Investigations of R. Mertens et al., and Brüggemann et al., with non-heated and pre-heated build platforms (RT, 200 °C, 400 °C) reveal for AA7075 a reduction of hot cracking and a significant change in crack-formation having increasing intervals between the cracks in conjunction with increasing platform temperature [40,41]. For some powder variants, both single and double scanning were performed. Each parameter study consisted of 35 specimens with a dimension of 10 × 12 × 10 mm³ (width, high, depth) which were built without support structure and with the same environmental condition. The scan strategy of all samples was meander shaped with a 67° rotation from layer to layer. To avoid hot cracks as well as to achieve high relative density, the parameter adaption was performed with a laser power (P_L_) between 290 W and 370 W, scan speed (v_s_) in the range of 600 mm/s and 1200 mm/s as well as a hatch distance (h_d_) between 0.08 mm and 0.14 mm. A constant layer thickness of 50 µm was applied. To reduce the required powder amount, a customized coater with a narrow slot leads to a powder layer of approximately 70 mm width. Before the laser beam process starts, the coater was directly filled with dried powder, enough for a complete build job.

### 2.3. Characterization of Powder and Specimens

The particle size distribution (PSD) of the starting powder, as well as after mechanical decoration without the addition of grain refiner (ball milling), was determined via laser diffraction (Mastersizer 2000, Malvern Instruments, UK). Investigations concerning powder flowability of virgin powder and AA7075 powder incorporated with TiC NPs were analyzed, employing a ring shear tester RST-XS.s (Schulze). Following ASTM D6773, a shear normal stress of 2500 Pa for the gauged bulk density (1.2 g/cm–1.5 g/cm^3^) was applied [42]. Dried powder with a relative moisture ≤5% was tested. Three measurements were performed with the small ring shear cell M1 (V = 31 cm^3^). The details on all experimental aspects of the DFSC analysis including temperature correction are published by Zhuravlev et al. [43].

To ascertain the relative density of the specimens, vertical cross-sections with an area of ≈1 cm^2^ for each sample were prepared. All specimens were mounted in epoxy resin for grinding with the automatic preparation system Hexamatic (Struers GmbH, Germany). The utilized method includes a material removal of at least 1 mm by a grindstone. Subsequently, fine grinding and polishing with a final silicon oxide polishing suspension (50 nm particles) for 15 min were employed to gain an appropriate surface quality for light microscopy. The stereology determination of density and detection of cracks was carried out using the digital microscope Keyence VHX-5000, equipped with the lens VH-Z100UR. All images were taken with a magnification of 200×. Hence, each pixel represents an area of 1.031 × 1.031° µm^2^. The determination of the part density was performed with software by comparing light and dark areas with a manual set threshold value. In this study, defects were identified as cracks if the perpendicular Feret’s diameter was at least four times the horizontal Feret’s diameter. This method is based on the evaluation criteria applied by Damon et al. [44]. For microstructure analysis, a Zeiss Ultra Plus scanning electron microscope (SEM) equipped with a Digiview 5 detector for electron backscatter diffraction (EBSD) and an OCTANE PRO detector for energy-dispersive X-ray spectroscopy (EDS) was used (both detectors: EDAX, AMATEK Inc., USA). A section of approximately 20 µm × 20 µm was prepared by focused ion beam (FIB) using SEM-FIB Zeiss Neon 40 at an acceleration voltage of 20 kV and a probe current of ~2 nA. Grain size analysis according to DIN EN ISO 643 [45] and optical microscopy (OM) were performed via a light microscope (Axiophot, Zeiss, Germany). Ground and subsequently electro-polished as well as etched (WECK reagent) surfaces (in building direction) of the cuboids were analyzed.

Measurements of the chemical composition were conducted with X-ray fluorescence (XRF) spectroscopy (S8 Tiger, Bruker AXS, USA) employing an X-ray tube of Rhodium with a Beryllium window at 4 kW. For each powder sample, 3 g were analyzed with a 34 mm aperture under negative Helium pressure. To analyze the quantity (ma%) of surface-decorated nanoparticles on the base material, the powder was directly measured, normalized and aluminum (Al) was introduced as a matrix.

Tensile tests were performed with tensile samples having an overall length of 26 mm, a width of grip section of 7 mm, and a thickness of 1.5 mm to determine the mechanical properties. Therefore, a servo-hydraulic test rig MTS 858 Table Top System and an extensometer model 632.29F-30 (both MTS Systems, USA) were applied. Each tensile test was displacement controlled using a traverse rate of 1.5 mm/min and in regular ambient conditions. A customized excel script was employed for the analysis and graphic editing of test results. Some tensile samples were heat-treated applying double ageing according to Sun et al. [46]: solution annealing at 470 °C for 1 h and subsequent quenched with water, then the 1st ageing at 110 °C for 5 h following the 2nd ageing employing 150 °C for 14 h.

## 3. Results and Discussion

### 3.1. Effect of Mechanical Decoration on Powder Properties

In Figure 1, the particle size distribution for both, starting powder as well as after ball milling (without added nanoparticles) is presented with logarithmic scaling versus the relative frequency. The particle size distribution of ball-milled powders was investigated to consider the influence of the used balls on the particles, respectively the particle size of the base material. This facilitates an evaluation of the effect of added TiC NPs and resulting powder characteristics, in particular concerning flowability. The slight decrease in the peak and the shift of the curve of the ball-milled powder towards smaller particle sizes indicates an increasing number of finer particles after ball milling and a quantitative reduction of particles with a size of about 35 µm and above. It is assumed that the addition of TiC NPs has only marginal effects on the change in particle size during the mechanical decoration procedure.

Figure 2a shows particles of the virgin AA7075 powder. A few satellites stick on the surface. Some severely deformed and non-circular particles after the mechanical decoration process are seen in Figure 2b,c. The surface as well as spaces between the powder particles exhibit attachments of larger quantities of significantly smaller particles, most likely the added nanoparticles. This can be seen more distinctly in the higher magnification (Figure 2c). Moreover, the surface structure (i.e., grains) of the AA7075 powder particles after the mechanical decoration process with 2.5 ma% TiC NPs appears less pronounced than in the virgin state. The presence of TiC was proved by EDX, deployed to the area marked by a dashed frame in Figure 2c.

The flowability factor ff_c_ of the virgin starting material (AA7075), ball-milled powder (AA7075/BM), and the modified powder variants with 0.5 ma% TiC NPs (AA7075/TiC/0.5) and 1.0 ma% TiC NPs (AA7075/TiC/1.0) are illustrated in Figure 3. Three measurements were conducted for each powder variant. From the consolidation stress and compressive strength determined with the ring shear tester, the respective flowability values are depicted. For the evaluation of the flowability, the ff_c_ and the classification following Jenike [47] is deployed. All powders in the test series were subjected to a vacuum drying process before the ring shear test to achieve a relative humidity of the powder of less than 5 % at ambient temperature (≈22 °C).

The highest flowability factor (ff_c_ = 14.6) and therefore the best flowability is determined for AA7075/TiC/1.0. When the powder is undergoing the mechanical decoration process without NPs (only ball milling), this powder has a lower flowability (average ff_c_ = 8.9) at the end of the process than at the beginning, i.e., as the starting powder (ff_c_ = 10.8). It has to be considered that the results after ball milling represent only the effect of the applied process-specified parameters (e.g., ball size, powder-to-ball weight ratio, filling level) on the AA7075 powder.

After mechanical decorating, respectively ball milling, some particles show (partially severe) deformations. Larger particles are crushed during the process and the proportion of small particles increases. Nevertheless, the majority of particles have a rounded shape. An increase in the fines fraction reduces the flowability of powders [48]. This is also seen for the ball-milled powder. The number of deformed particles and the increase in the proportion of finer particles derived from the starting powder do not reduce the flowability if nanoparticles are added. Therefore, an improved flowability results from the addition of nanoparticles. It appears that the threshold for the quantity of TiC NPs to compensate for the negative effects caused by ball milling is below 0.5 ma%.

Figure 4 highlights the onset temperature of solidification of five samples for each of the three investigated powder variants and repeated experiments as a function of the cooling rate using a logarithmic scale. The three powder variants with different symbols are AA7075 (black-bordered rhombus), AA7075/TiC/1.0 (green triangles), and AA7075/TiC/2.5 (orange dots). The onset temperatures are scattered, while the scatter for the three powder variants is considerably different. The solidification onset temperatures are averaged for each variant considering all cooling rates given as a horizontal straight line. On the left side, the corresponding mean values and the mean dispersion are given.

As seen in Figure 4, the undercooling decreases for all cooling rates after the modification with TiC nanoparticles. The undercooling temperature and the scatter for the three measured powder variants are listed in Table 2. It has to be noted that for the powder variant AA7075/TiC/0.5 another batch of starting material with minor deviation in the Zn and Mg content was used and hence a higher solidus temperature is expected. The undercooling calculated (Table 2) is the difference between the solidus temperature taken from the literature [49] and the measured solidification onset temperature from DFSC. The solidus temperature is not available specific to the AA7075 base powder used for the investigation here and might complicate the comparison slightly. However, the general trend in terms of undercooling as well as its scatter is obvious.

In addition to the chemical composition of the base material AA7075 (Table 1), XRF was also applied to determine the chemical composition of the powders modified with TiC nanoparticles. Table 3 summarizes the concentration of Al, the main alloying elements of AA7075: Zn, Mg, and Cu as well as Si and Ti of the powders investigated. To be able to exclude a possible loss of TiC before the additive manufacturing process, virgin starting material AA7075, the surface-modified powders AA7075 with 1.0 ma% TiC NPs after mechanical decoration (AA7075/TiC/1.0), after sieving (SI), and after vacuum drying (VD) are measured by XRF to trace the Ti content along with the entire powder modification and preparation route. Moreover, the same measurement methodology is used to determine the chemical composition of powder particles carried by the shielding gas and deposited in front of the gas outflow (CP), oversized particles after sifting of used powder (OP), and as-built parts (BS).

It is seen that the Ti content of AA7075/TiC/1.0 increases compared with AA7075 in the same quantity as TiC NPs were added before the mechanical decoration. Even after sieving and vacuum drying, the Ti content remains almost constant within the limits of the measurement accuracy. Both the gas-carried powder particles and the oversize particles exhibit an increased Ti content. Most of the oversize particles are quite likely weld spatter. The low content of Ti in the additively manufactured specimens additionally implies the loss of TiC NPs during the PBF-LB/M process.

The mechanical decoration process shown here seems to be a suitable method for powder modification with TiC NPs regarding powder flowability and affecting the undercooling and its scatter. Additionally, it is clear from the results of the XRF analysis that the amount of TiC NPs added, and thus the nuclei that are potentially available to form the grain refinement, are not significantly affected by the powder handling. Therefore, the same amount of TiC NPs added is also be assumed in situ, during PBF-LB/M. Besides the positive influence on the flowability of the powder, the mechanical decoration is a technically easy-to-implement process for the surface application of nanoparticles to aluminum powder.

### 3.2. After PBF-LB/M of Modified Powder

Figure 5 shows the resulting relative density (green squares) and associated scale on the left as well as the crack density (red rhombus) and the corresponding scale on the right site as a function of the occurring volumetric energy input (E_V_). The range of the E_V_ within the deployed 35 parameter ranges between 34.5 J/mm^3^ and 154.2 J/mm^3^ with non-uniform distribution in between. Each symbol represents a result based on the microscopic analysis of the cross-section to determine the ratio of material and lack of material as well as counting the observed cracks on the investigated area.

With increasing energy input, a tendency of decreasing crack density can be perceived. Of 29 additively manufactured samples with the applied parameter settings having a volumetric energy input of 50 J/mm^3^ and higher, 28 samples exhibit a density of >97.5%. The influence on the microstructure of different parameter settings deployed is illustrated in Figure 6. With the lowest volumetric energy input of 34.5 J/mm^3^ in the parameter study, a large number of voids occurs, which results in a relative density of only 83.1% (Figure 6a). Figure 6b,c show the cross-section of specimens manufactured with an energy input of 66.7 J/mm^3^, respective 69.0 J/mm^3^. A distinct increase in dense material can be noted, as well as occasional hot tears. The sample with the highest relative density and without apparent cracks is depicted in Figure 6d. In Figure 6e, rounder pores and thus a reduction of the relative density can be seen. For this cuboid specimen, the highest energy input of 154.2 J/mm^3^ was set.

It is ascertained that a lower, as well as an upper, energy input results in an insufficient part density and the occurrence of cracks. However, with the parameter setting listed in No 7 (Table 4), in conjunction with the modified powder variant AA7075/TiC/2.5, crack-free specimens with a density of 99.9% were achieved.

The occurrence and extend of grain refinement were determined by SEM-EBSD examination and are illustrated in Figure 7. Each micrograph shown represents a section of 200 × 200 µm^2^. The microstructure of Figure 7a–c originates from samples that were built with the same parameter set (No 1–3, Table 4) based on the best result, i.e., highest part density and lowest crack density for powder variant 7075/TiC/1.0. In Figure 7d, the microstructure of the sample with the best results (No 7, Table 4) made of powder variant 7075/TiC/2.5 is seen.

Whereas columnar grains are visible in AA7075 (Figure 7a), the proportion of non-equiaxial grains decreases significantly with TiC NP modified powders. The microstructure of AA7075/TiC/1.0, seen in Figure 7c, consists solely of equiaxed grains. With an increased amount of TiC NPs up to 2.5 ma%, seen in Figure 7d for the powder variant AA7075/TiC/2.5, the grains become smaller.

In Figure 8, the average grain sizes for different powder variants depending on the PBF-LB/M conditions are depicted. The results shown for AA7075, AA7075/TiC/0.5, and AA7075/TiC/1.0 are from specimens built with the parameter set that achieved the highest density and the lowest crack number gained for AA7075/TiC/1.0 (No 3, Table 4). Furthermore, the specimen built with the same parameter set but a different scanning number was analyzed. The samples made of AA7075/TiC/1.75 and AA7075/TiC/2.5 were manufactured with the best parameter set, i.e., the process parameters to obtain the lowest porosity and the lowest crack density at the same time. Due to the declining impact of the scanning number on the grain size with an increasing proportion of TiC NPs and the aspect of the (economical) application and manufacturing of the modified material in the industrial field, only single-exposed specimens were built and investigated for the powder variants AA7075/TiC/1.75 and AA7075/TiC/2.5.

Nevertheless, it should be noted that for variant AA7075/TiC/1.0, the grain sizes after single scanning (4.8 µm) and double scanning (4.9 µm) are close to each other, but the reduction in the number of cracks is notable in double-exposed specimens. The microstructure of the unmodified AA7075, processed with the here stated process condition, leads to grains with an average diameter of up to 44.7 µm. With AA7075/TiC/2.5 using a parameter set according to No 7 listed in Table 4, an average grain size of 1.7 µm was attained. It should be noted that all measured values are related to the mentioned conditions here. Based on the grain structure (Figure 7) and the measured grain diameters (Figure 8), a decrease in grain size and a change towards an equiaxed shape of the grains are observed with the increasing addition of TiC NP to the starting powder.

Figure 9 displays micrograph sections with a size of 7.5 × 7.5 mm^2^. The images in each column result from additive processing of different powder variants and applying single exposure with the same PBF-LB/M parameter set. The specimens whose cross-sections are depicted in the left column (Figure 9a,c,e,g,i) were manufactured with the parameters shown under No 7 (Table 4). This is the parameter set with the highest density and the lowest number of cracks, respectively crack-free results obtained with the AA7075/TiC/2.5 powder variant. For the samples and their micrographs displayed in the right column, the parameter set with an E_V_ of 137.5 J/mm^3^ (P_L_ = 330 W; v_s_ = 600 mm/s; h_d_ = 0.08 mm) was applied. Line by line from top to bottom, the following powder variants were used for the samples: AA7075, AA7075/TiC/0.5, AA7075/TiC/1.0, AA7075/TiC/1.75, AA7075/TiC/2.5.

For AA7075 (Figure 9a,b), cracks and defects occur extensively regardless of the parameter settings applied for the PBF-LB/M. As already shown in Figure 6, a higher porosity is observed with higher energy inputs. The same was observed in the right column, particularly for the non-modified AA7075 (Figure 9b) and the TiC NP modified powder variants (Figure 9d,f,h) with decreasing extent. No significant differences in either the relative part density or the crack density were observed for the samples made of AA7075/TiC/2.5 but processed with different parameter settings (Figure 9i,j). The vertical cracks (hot cracks) propagated along with the build direction, getting larger and having similar distances to each other (kind of pattern) at some specimens built using TiC modified powder (Figure 9e,f,g). Such an appearance is known in additive processing of AA7075 using higher build platform temperatures [41]. A higher addition of TiC NP does not equally imply a reduction of cracks at the same process conditions. This is seen in Figure 9e for AA7075/TiC/1.0 where the microstructure exhibits no visual discernible hot tears compared to AA7075/TiC/0.5 but also AA7075/TiC/1.75.

Both the investigation of microstructure by digital microscopy as well as SEM demonstrate a strong dependence of the applied parameter settings as well as the material modification on the crack number and relative part density at the via laser beam melting manufactured samples. However, it is of great importance that the parameter setting is suitable for the respective and ultimately for the powder variant AA7075/TiC/2.5 to achieve material properties that are sufficient for technical use. Successful processing of AA7075 with TiC NPs deposited by mechanical decoration requires the addition of a sufficient amount of TiC NPs. In addition, it was shown that grain refinement on its own is not sufficient to eliminate hot cracks. There also seems to be a maximum size (≈2 µm) of grains that must be undercut here. Another aspect is the difference between the theoretical number of NPs and the detected number of grains. According to Greer et al. [23] at best 1%, of the added particles lead to nucleation. To estimate the theoretically available nuclei at AA7075/TiC/2.5, a simple calculation based on an AA7075 particle diameter of 40 µm (measured mean is 40.8 µm), a TiC particle size of 40 nm (measured mean is 40.1 nm), 2.5 ma% (1.44 vol%) enclosed TiC NPs (density: 4.9 g/cm^3^) related to each particle of AA7075 (density: 2.8 g/cm^3^), and the resulting grain size of 1.7 µm is used. The result implies for the condition mentioned and separately present nanoparticles (not agglomerated), that approximately 0.1% of the added nanoparticles act as nuclei.

Figure 10 demonstrates the microstructure revealed by SEM-FiB cutting and EDX analysis of two areas with different sized grain structures. The additive manufactured cuboid specimen using parameter set No 7 according to Table 4 consisting of AA7075/TiC/2.5 with a ground and polished surface was SEM-FiB-cut. The cutting plane shown in Figure 10a,b is perpendicular to the building direction. The non-columnar grains appear in different shades of grey. Lighter regions are discernible at the grain boundaries and indicate a different chemical composition (segregation) from that of the surrounding grains. It is assumed that local segregation and enrichment of low-melting alloying elements, e.g., Zn and Mg at the grain boundaries occur during the formation of the grains. Figure 10b exhibits a finer structured area in size of approximately 4 µm by 2 µm in size. This indicates an accumulation of finer particles, e.g., TiC NPs. The chemical composition determined by EDX via spot analysis (ROI 2), depicted in Figure 10c, shows higher intensities for the characteristic energy of X-ray of Ti compared to the spectrum derived from ROI 1. This confirms the assumption of having a gathering of TiC NPs.

The characterization of the mechanical properties was conducted via tensile tests. For this purpose, tensile specimens (insert in Figure 11a) were prepared by spark eroding from an additively manufactured block. The tensile direction is the same as the layer building direction. In Figure 11a, three stress-strain curves are shown for the as-built condition (green) and heat-treated condition (red) for AA7075/TiC/2.5. The resulting behavior in tensile tests concerning the as-built (green) and heat-treated condition (red) is plotted in the stress-strain diagram seen in Figure 11b. Four samples for each variant were tested here. The parameter set listed in No 07 (Table 4) was used for the samples made of AA7075/TiC/2.5 and the parameters shown in No 01 (Table 4) were used for the AA7075/TiC/1.75 tensile specimens.

A relatively large scattering range can be seen for almost all tested powder variants and conditions. This concerns in particular the elongation at break. Although it was not possible to achieve a complete absence of cracks in the parameter analysis applying AA7075/TiC/1.75, a higher mean yield strength (+17.5%), higher mean ultimate tensile strengths (UTS) (+15.5 %), and lower mean elongation at break (−21.1%) are obtained compared with AA7075/TiC/2.5. A UTS close to 330 MPa in the as-built condition was achieved with AA7075/TiC/1.75 as well as AA7075/TiC/2.5.

In the heat-treated condition, the yield strength could be increased on average by more than 50% for AA7075/TiC/2.5 and more than 56 % for AA7075/TiC/1.75. The UTS was increased by 32% and by 41% after heat treatment, respectively. After heat treatment, the highest UTS of all specimens is 465 MPa and was attained with AA7075 modified with 1.75 ma% TiC NPs. For the same material, in addition to the expected improvement in yield strength and UTS after heat treatment, the rise in elongation at break is conspicuous. The best tensile test results achieved with a heat-treated AA7075/TiC/2.5 tensile sample is 344 MPa for the yield strength, a UTS of 384 MPa, and 4% elongation at break.

Apart from the (sporadic) presence of cracks in AA7075/TiC/1.75, occasional pores or defects are apparent on the surface of the tensile specimens, from which a reduction in the relative part density can be inferred. This is due to the use of the same meandering scanning strategy for the (smaller) samples for determining the part density. The porosity and defect frequency increase in the middle of the blocks used for the tensile specimen for both, AA7075/TiC/1.75 and AA7075/TiC/2.5. Nevertheless, the AA7075/TiC/2.5 tensile samples do not have any visible surface imperfections. Furthermore, it is assumed that the Portevin-Le-Chatelier effect [50] is the reason for the distinct zigzag stress-strain curves which appears more intensively at AA7075/TiC/2.5.

One reason for the decrease in strength likely comes with an increasing TiC NP content in AA7075. Accumulation of TiC NP agglomeration as seen in Figure 10a,b could lead to a vulnerability of the matrix to applied loads. A bunch of TiC NPs behave like a defect since there is no prevailing strong bond between the NPs. For AA7075/TiC/1.75, 1 vol% of NPs are present in the powder. Even after PBF-LB/M and the loss of NPs by melt pool ejection, a relatively large proportion of NP is not acting as nuclei and remain in the matrix. For the used tensile stress, the low number of hot cracks (AA7075/TiC/1.75) has no significant influence on the strength since the crack extension and tensile direction are the same.

## 4. Conclusions

This study demonstrate that high-strength aluminum alloys can be processed into crack-free and dense parts by using PBF-LB/M with the addition of 2.5 ma% TiC NPs. With this modified material and the process parameters: P_L_ = 350 W, v_s_ = 750 mm/s, and h_d_ = 0.10 mm, relative densities of >99.9 % with decent strength and ductility were achieved. The requirements for additive manufacturing of AA7075 with TiC NP and results from the investigations are summarized as follows:▪The amount of TiC that must be added to AA7075 for significant crack reduction and crack prevention over a wide range of PBF-LB/M parameter variations must be a minimum of 1.75 ma%.▪For modified AA7075 with TiC nanoparticles, following effects occur:▪Reduction of the solidification undercooling and its scatter.▪Nucleation of the additional grains and substantial reduction of the resulting grain size.▪Agglomeration of excess TiC NPs, which potentially introduce structural defects.▪So far not specified influence on the subsequent result of heat treatment by various structures (grain size and additional nucleation sites for secondary precipitation).
▪With the addition of TiC, for each powder variation, appropriate parameter settings for the PBF-LB/M process must be applied.▪The mechanical decoration is a suitable method in terms of:▪Scalability industrial application.▪Increasing the flowability of the metal powder.▪Sufficient dispersion of TiC NPs on the powder surface and bonding between the nanoparticle and AA7075 powder particle.



## Figures and Tables

**Figure 1 materials-14-07190-f001:**
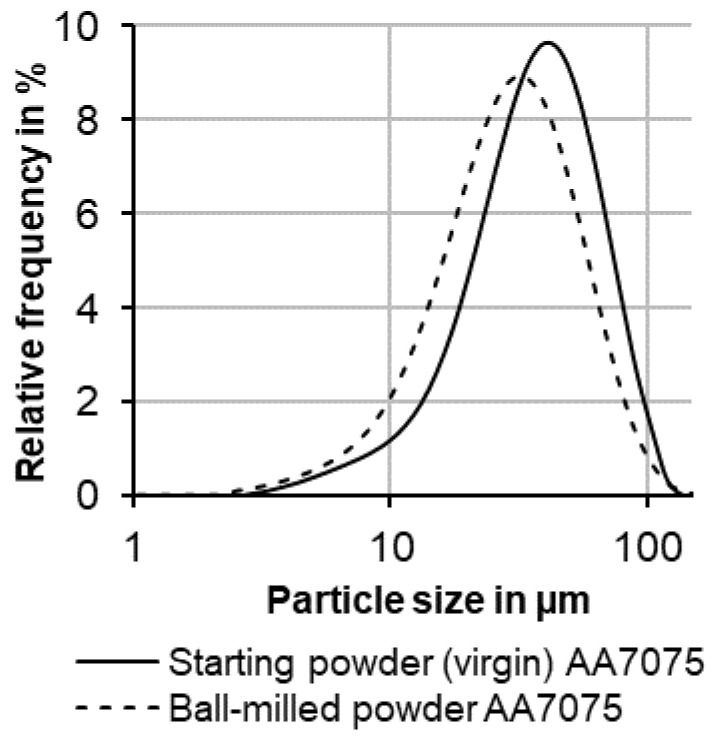
Powder particle size distribution before and after the ball milling process (without TiC NPs).

**Figure 2 materials-14-07190-f002:**
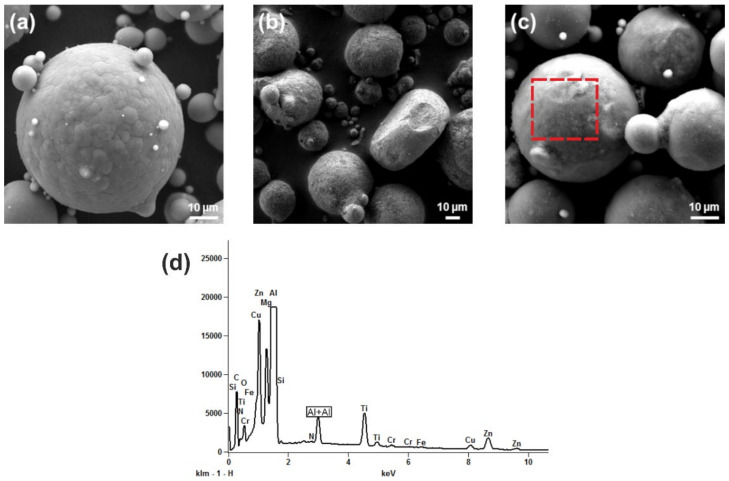
SEM-SE images of powder particles of (**a**) virgin AA7075 powder, (**b**) deformed powder particles of AA7075 powder after mechanical decoration with the addition of 2.5 ma% TiC nanoparticles, (**c**) same condition as in (**b**) at a higher magnification, and (**d**) EDX spectra of the dashed outline section highlighted in (**c**).

**Figure 3 materials-14-07190-f003:**
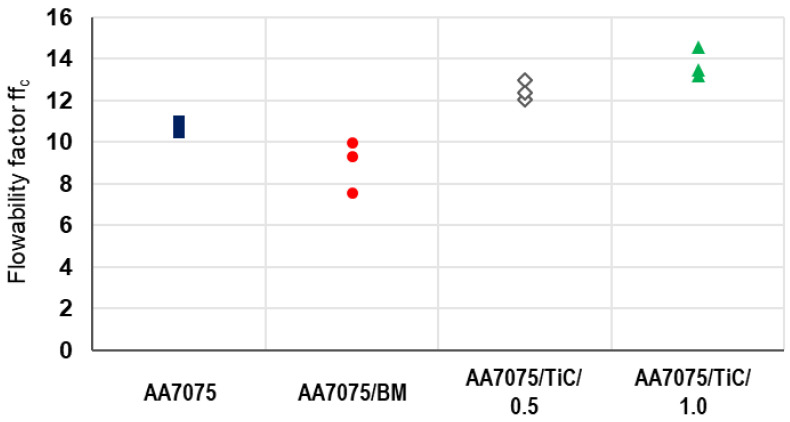
Flowability factor ff_c_ for different powder variants: starting powder AA7075 before and after ball milling (BM), but also mechanically decorated AA7075 powder with TiC NPs (0.5 ma% and 1.0 ma%).

**Figure 4 materials-14-07190-f004:**
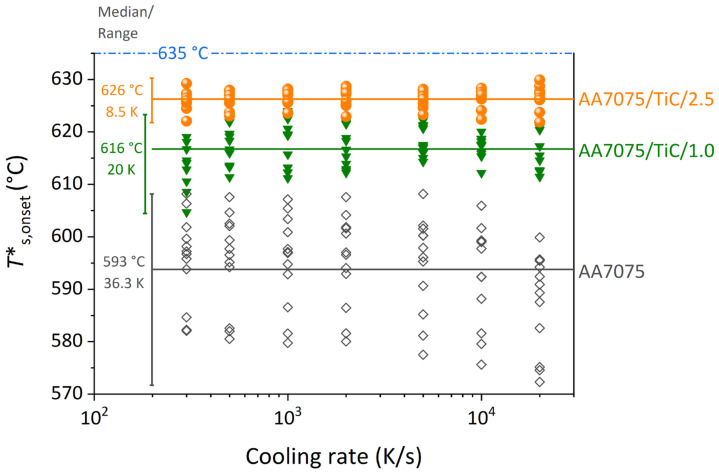
Solidification onset temperatures from DFSC for different powder variants as a function of the cooling rate in the range of 100 K/s–20,000 K/s.

**Figure 5 materials-14-07190-f005:**
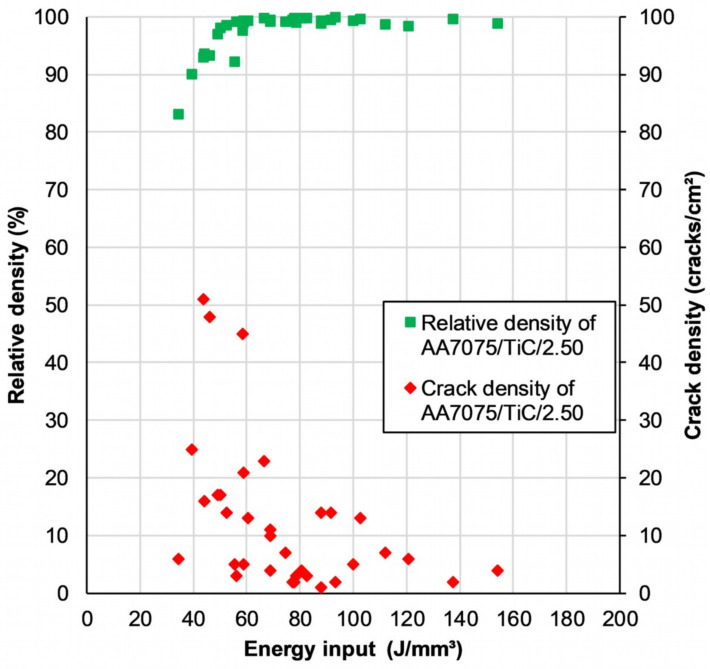
Correlation of the relative sample density and the crack density using powder variant AA7075/TiC/2.5 with the induced volumetric energy input from PBF-LB/M.

**Figure 6 materials-14-07190-f006:**
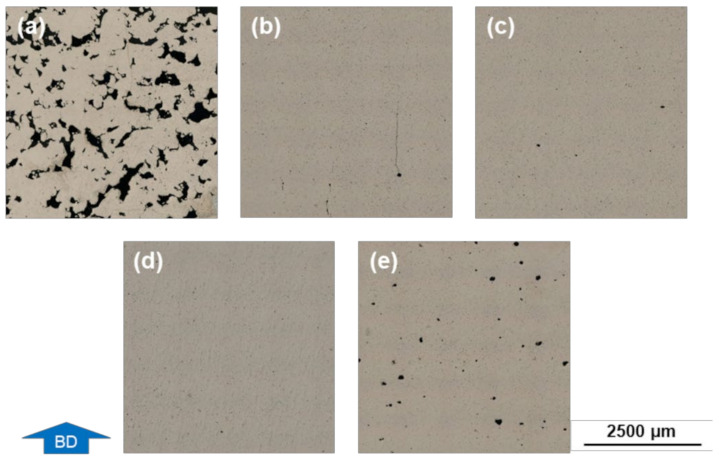
Micrographs of cross-sectional cuboid samples taken with a digital microscope. PBF-LB/M samples made of AA7075/TiC/2.5 with different process parameters and resulting volumetric energy input: (**a**) E_V_ = 34.5 J/mm^3^; (**b**) E_V_ = 66.7 J/mm^3^ (**c**) E_V_ = 69.0 J/mm^3^; (**d**) E_V_ = 93.3 J/mm^3^; (**e**) E_V_ = 154.2 J/mm^3^.

**Figure 7 materials-14-07190-f007:**
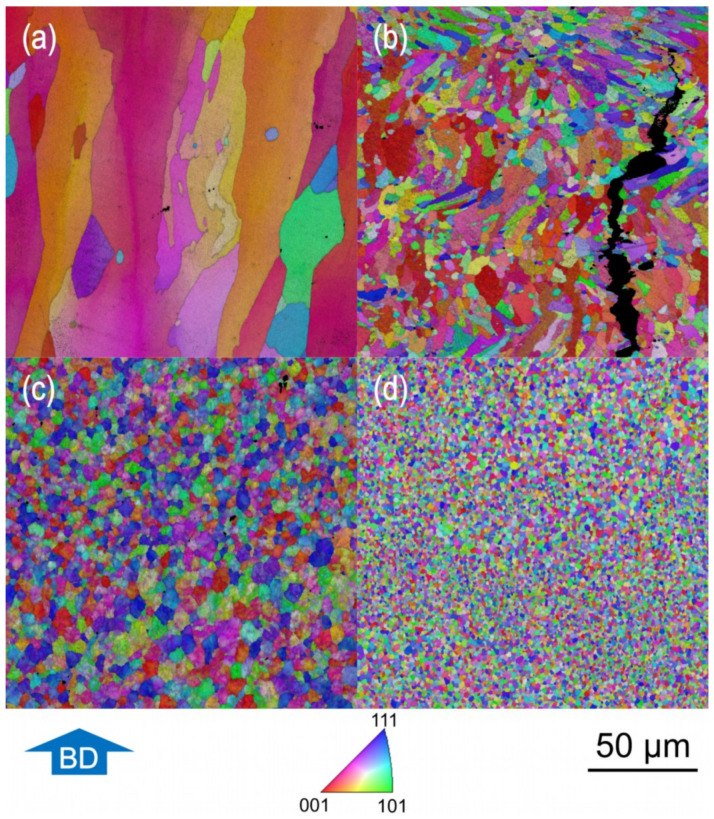
SEM-EBSD images indicating the grain structure of samples achieved with different powder variants: (**a**) AA7075, (**b**) AA7075/TiC/0.5, (**c**) AA7075/TiC/1.0, and (**d**) AA7075/TiC/2.5.

**Figure 8 materials-14-07190-f008:**
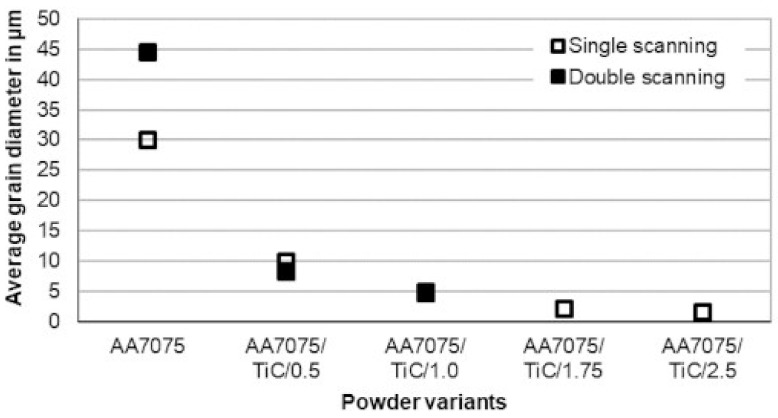
Average grain sizes affected by different powder variants, scanning numbers, and parameter settings.

**Figure 9 materials-14-07190-f009:**
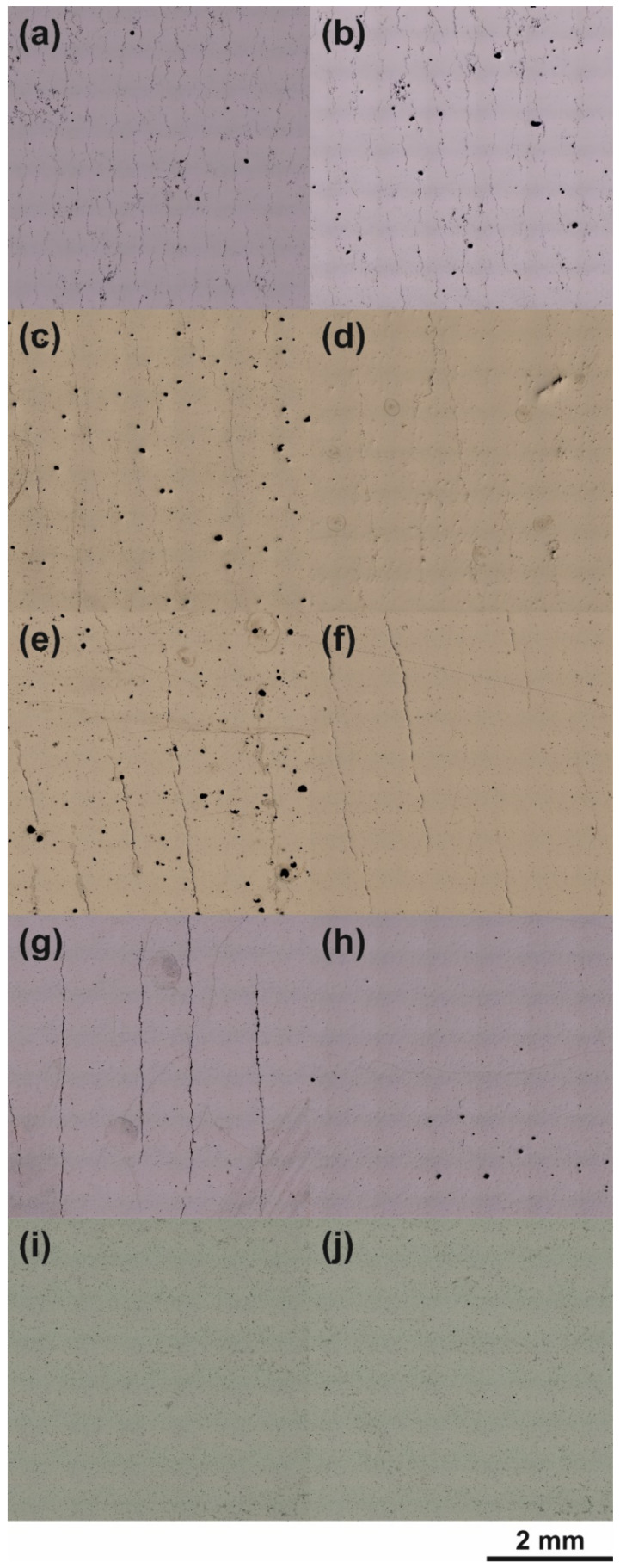
Micrographs of samples derived from the different powder variants, manufactured with the same parameters (one parameter set per column) via PBF-LB/M: (**a**,**b**) AA7075, (**c**,**d**) AA7075/TiC/0.5, (**e**,**f**) AA7075/TiC/1.0, (**g**,**h**) AA7075/TiC/1.75, (**i**,**j**) AA7075/TiC/2.5.

**Figure 10 materials-14-07190-f010:**
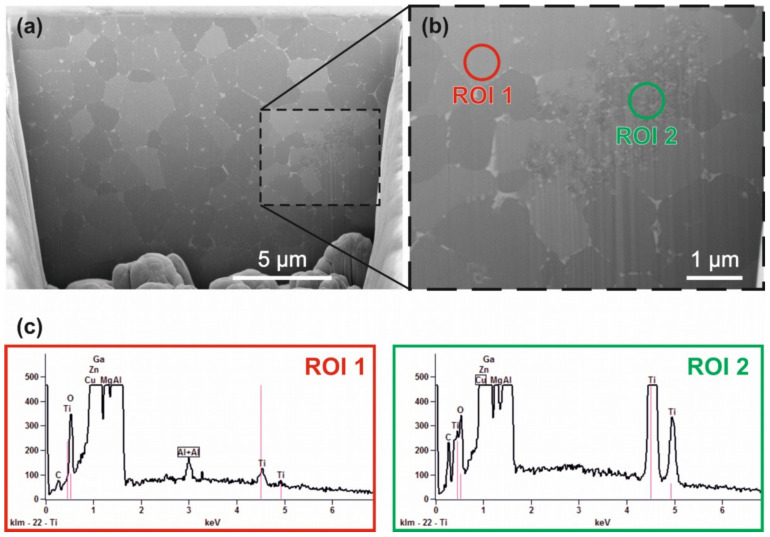
SEM-FiB sample of an as-built specimen made of AA7075/TiC/2.5 giving an insight into the microstructure. (**a**) SEM-SE image of the cut surface in total, (**b**) magnification of the dashed outlined area, (**c**) EDX spectra of the two regions of interest (ROI), indicating an enrichment in Ti within the finer structured region seen in image (**b**).

**Figure 11 materials-14-07190-f011:**
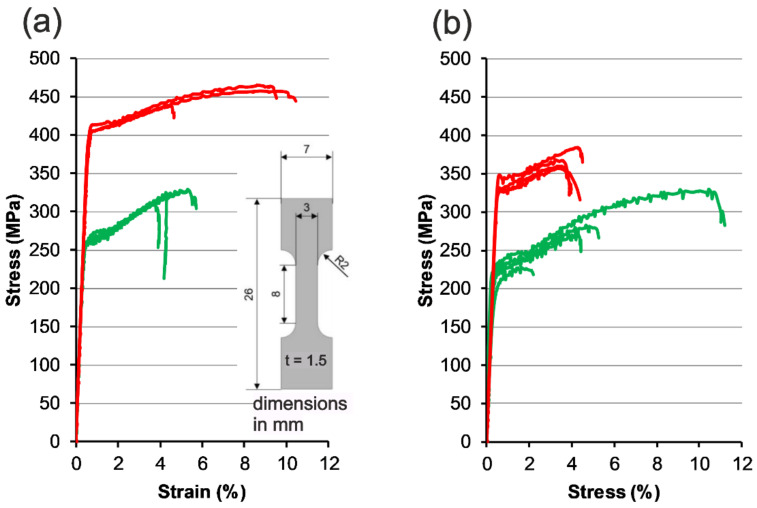
Stress-strain curves of as-built (green) and heat-treated (red) tensile specimens manufactured with the powder variant (**a**) AA7075/TiC/1.75 and (**b**) AA7075/TiC/2.5.

**Table 1 materials-14-07190-t001:** Mass fractions of alloying elements of the starting powder AA7075 measured via energy-dispersive X-ray spectroscopy.

Material	Elemental Composition in ma%						
	Al	Zn	Mg	Cu	Si	Fe	Ti
AA7075	91.70	5.04	1.26	1.46	0.05	0.11	0.03

**Table 2 materials-14-07190-t002:** Undercooling during solidification and its scatter for various powder variants.

Powder Variant	Undercooling [ΔT_U_] in K	Scatter in K
AA7075	42	36
AA7075/TiC/0.5	14 [43]	11 [43]
AA7075/TiC/1.0	19	20
AA7075/TiC/2.5	9	9

**Table 3 materials-14-07190-t003:** Elemental composition for powder, powder derivate, and part manufactured by PBF-LB/M determined by energy-dispersive X-ray spectroscopy.

	Origin	Condition	Elemental Composition in ma%					
			Al	Zn	Mg	Cu	Si	Ti
**Powder modification and preparation for PBF-LB/M**	AA7075/TiC/1.0	Modified	90.60	5.07	1.29	1.50	0.06	1.03
		SI	90.60	5.10	1.20	1.47	0.07	0.99
		VD	90.40	5.18	1.28	1.47	0.10	1.05
**After PBF-LB/M**	AA7075/TiC/1.0	CP	94.40	1.21	1.13	1.51	0.12	1.21
		OP	94.40	1.36	1.00	1.46	0.11	1.55
		BS	91.30	3.85	1.35	1.57	0.28	0.77

**Table 4 materials-14-07190-t004:** Resulting crack density and part density for various powder variants and different process parameter settings of PBF-LB/M.

No	Powder Variant	Figure	Laser Power [P_L_] in W	Scan Speed [v_s_] in m/s	Hatch Distance [h_d_] in mm	Volumetric Energy Input [E_V_] in J/mm^3^	Relative Part Density in %	Crack Density in Cracks/cm^2^ *	Remark
1	AA7075	Figure 7a	370	900	0.08	102.8	99.50	154	Single-exposed
2	7075/TiC/0.5	Figure 7b	370	900	0.08	102.8	-	-	Single-exposed
3	7075/TiC/1.0	Figure 7c	370	900	0.08	102.8	99.33	2	Double-exposed
4	7075/TiC/2.5	Figure 6a	290	1200	0.14	34.5	83.1	6	Lowest E_V_ in the parameter study
5	7075/TiC/2.5	Figure 6b	350	1050	0.10	66.7 **	99.8	23	
6	7075/TiC/2.5	Figure 6c	290	600	0.14	69.0	99.4	10	
7	7075/TiC/2.5	Figure 6d and Figure 7d	350	750	0.10	93.3	99.9	2	Parameter setting to achieve the highest density for material variation AA7075/TiC/2.5
8	7075/TiC/2.5	Figure 6e	370	600	0.08	154.2	98.8	4	Highest E_V_ in the parameter study

* It should be noted that a low crack density is not synonymous with a high part density, and a material defect detected and identified as a crack can also be a (small) vertical elongated pore. ** A sample built with different parameter settings, but with the same E_V_, exhibits a density of 99.9 % and 8 (visual perceptible) cracks per cm^2^.

## Data Availability

The data presented in this study are available on request from the corresponding author.

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
