# Peer review of "Requirements for Processing High-Strength AlZnMgCu Alloys with PBF-LB/M to Achieve Crack-Free and Dense Parts"

_materials, 2021, doi:10.3390/ma14237190_

Round 1
Reviewer 1 Report
The manuscript “Requirements for successfully processing high-strength AlZnMgCu alloys with PBF-LB/M” by Heiland and the team discusses the laser powder bed fusion of novel aluminum alloy 7075 with 2.5 % titanium carbide nanoparticles. Heat and mechanical tests were carried out on the printed samples. The results are novel and the article is suitable to Materials journal. Nevertheless, the article needs significant English improvement throughout the manuscript before further consideration. Moreover, the manuscript could be considered for publication after the following mandatory modifications.
1. The title of the manuscript sounds very vague. Make it to be more specific.
Abstract
2. What means powder bed fusion? Is it laser powder bed fusion? If not, it is recommended to define the term.
3. Are the authors sure that “Furthermore, for the first time, it is demonstrated that not all nanoparticles used remain in the melt pool during PBF-LB/M”? It is important to mention a few studies before claiming this novelty.
Introduction
4. There is no term such as “powder bed fusion of metals with a laser beam.” Instead, the proper term is laser powder bed fusion (LPBF). Please, correct accordingly.
5. Reference citing trend is not homogenous throughout the introduction section. Make it uniform.
6. The literature review lacks significant contributions such as problems with Metal matrix composites machining and achievement of homogenous distribution of dispersed phase within the matrix. This reviewer suggests to authors have a look to the following references:
DOI: 10.3390/ma13112593
DOI: 10.1080/10910344.2012.747897
7. The introduction could be significantly improved by including a section related to modelling that is a relevant tool besides experimentations. To identify, one may have a look at the following articles:
DOI: 10.3390/ma14164733
Materials and Methods
8. Explain in detail how the number of balls was selected for ball milling to achieve homogenous mixing of the dispersed phase in the matrix.
9. Were the samples printed using supports or without supports?
10. What scan strategy was implemented throughout the printing?
11. No information regarding the substrate’s material is available. Please, complete.
Results and Discussions
12. How the flowability of the powder particles was determined? Explain the setup in detail.
13. Line 350: Error! Reference source not found. Please, correct this error.
14. The conclusion section should be redefined and findings should be provided in more detail.
Author Response
Dear reviewer,
Thank you for your assessment of the manuscript and your recommendations for improvement. All comments and the subsequent need for change were discussed constructively by the authors.
Remark 1: The title of the manuscript sounds very vague. Make it to be more specific.
Statement 1: The heading is adapted and represent now more specific the content of the research work which is descript in the manuscript.
Remark 2: What means powder bed fusion? Is it laser powder bed fusion? If not, it is recommended to define the term.
Statement 2: The text passage (line 39) was changed and the additive manufacturing process is named according ISO/ASTM 52911-1:2019(en) Additive manufacturing – Design – Part 1: Laser-based bed fusion of metals.
Remark 3: Are the authors sure that “Furthermore, for the first time, it is demonstrated that not all nanoparticles used remain in the melt pool during PBF-LB/M”? It is important to mention a few studies before claiming this novelty.
Statement 3: If we could mention and refer to other studies, our finding would not be novel. But we are agreeing to you, to be safe, we omit the text part "for the first time".
Remark 4: There is no term such as “powder bed fusion of metals with a laser beam.” Instead, the proper term is laser powder bed fusion (LPBF). Please, correct accordingly.
Statement 4: Please see statement 2. Unfortunately, laser powder bed fusion (LPBF) is not the international and technical (neutral) correct term.
Remark 5: Reference citing trend is not homogenous throughout the introduction section.
Statement 5: For all citations, it is apparent where the stated information given comes from.
Remark 6: The literature review lacks significant contributions such as problems with Metal matrix composites machining and achievement of homogenous distribution of dispersed phase within the matrix. This reviewer suggests to authors have a look to the following references:
Statement 6: Concerning your literature proposal
10.3390/ma13112593; Metal Matrix Composites Synthesized by Laser-Melting Deposition: A Review”
The following investigation, named in the recommended review article, was added (line 90 - 93):
https://doi.org/10.1016/j.jmapro.2020.02.036; Effect of TiB2 content on microstructural features and hardness of TiB2/AA7075 composites manufactured by LMD by Lei, Zhenglong et al.
Concerning your literature proposal
10.1080/10910344.2012.747897; MACHINING OF MMCs: A REVIEW
This publication covers a wide range of MMCs and no specific aspect that can be brought into scientifically meaningful agreement with this manuscript. Moreover, the review does not address additive manufacturing.
Remark 7: The introduction could be significantly improved by including a section related to modelling that is a relevant tool besides experimentations. To identify, one may have a look at the following articles
Statement 7: The investigations carried out here intentionally do not contain modelling. Based on the findings gained, material factors and the influence of processing parameters may be determined via modelling in the further course of the project.
Remark 8: Explain in detail how the number of balls was selected for ball milling to achieve homogenous mixing of the dispersed phase in the matrix.
Statement 8: The number of balls was selected according to the requirements of the manufacturer of the rotation mill.
Remark 9: Were the samples printed using supports or without supports?
Statement 9: Added in line 162. The samples were built without support structure.
Remark 10: What scan strategy was implemented throughout the printing?
Statement 10: Detailed information in line 162/163
Remark 11: No information regarding the substrate’s material is available. Please, complete.
Statement: We conducted an optical emission spectroscopy analysis of the build tray. The chemical composition cannot allocate to an aluminum alloy listed in DIN EN 573-3:2019-10. In line 154, it is now mentioned, that the build tray is made of an aluminum alloy of the 5xxx series.
Remark 12: How was the flowability of the powder particles determined? Explain the setup in detail.
Statement 12: Please see line 177 – 182
Remark 13: Line 350: Error! Reference source not found.
Statement 13: Line 350 is an empty line!?
Remark 14: The conclusion section should be redefined and findings should be provided in more detail.
Statement 14: Agree! Please see revision.
Best regards,
Steffen Heiland

Reviewer 2 Report
The study “Requirements for successfully processing high-strength AlZnMgCu alloys with PBF-LB/M” is devoted to an application of AA7075 alloy in additive manufacturing. The paper corresponds well to the journal scope and well designed. The results are novel and reliable. However from the reviewer point of wiew the current research cannot be accepted in present state. Despite the good quality of results and presented information the conclusion section is reported in a too superficial manner and must be reformulated for being more concrete. It is not acceptable to use such formulas as e.g. “The amount of TiC added should be sufficiently large but as few as possible”. The conclusions should provide a concrete results and concrete values of achieved parameters. After the mentioned modification I would recommend the current research for publishing in Materials.
Author Response
Dear reviewer,
Thank you for your assessment of the manuscript and your recommendations for improvement. All comments and the subsequent need for change were discussed constructively by the authors.
Remark: Conclusion section is reported in a too superficial manner and must be reformulated for being more concrete.
Statement: Agree! Please see revision.
Best regards,
Steffen Heiland

Reviewer 3 Report
This is very interesting and well writen paper and it is worth of publishing in this Journal after minor corrections.
Abstract is very long with many unneeded facts. It requires a shortening and clearlier presentation of the aim of investigation.
Authors should give EDX analysis of the powder particles shown in Figure 2 to prove the presence of Ti.
Line 350, page 10: Error source not found. Please correct it.
Author Response
Dear reviewer,
Thank you for your assessment of the manuscript and your recommendations for improvement. All comments and the subsequent need for change were discussed constructively by the authors.
Remark 1: Abstract is very long with many unneeded facts.
Statement 1: Unnecessary information are eliminated.
Remark 2: EDX analysis of the powder particles shown in Figure 2 to prove the presence of Ti.
Statement 2: Revised.
Remark 3: Line 350, page 10: Error source not found
Statement 3: Line 350 is an empty line!?
Best regards,
Steffen Heiland
